# A Comparative Study of Three Dementia Screening Instruments (CSI-D, CMMSE, and ECAQ) in a Multi-Ethnic Asian Population

**DOI:** 10.3390/healthcare12030410

**Published:** 2024-02-05

**Authors:** Narayanaswamy Venketasubramanian

**Affiliations:** Raffles Neuroscience Centre, Raffles Hospital, Singapore 188770, Singapore; drnvramani@gmail.com; Tel.: +65-6311-1111

**Keywords:** cognitive screening, dementia, multi-ethnic, CSI-D, C MMSE, ECAQ

## Abstract

Background—There is no validated dementia screening tool for multi-ethnic Asian populations comprising Chinese, Malays, and Indians. This study aimed to establish the psychometric properties of the Community Screening Instrument for Dementia (CSI-D), Chinese Mini Mental State Examination (CMMSE), and Elderly Cognitive Assessment Questionnaire (ECAQ) in multi-ethnic Singapore. Methods—Participants were randomly drawn from a community-based neurological diseases study of older adults conducted in Singapore, with oversampling to allow similar subject numbers from the three major ethnicities: Chinese, Malay, and Indian. The CSI-D, CMMSE, and ECAQ were administered by trained research nurses using a standardised translated questionnaire in a language the participant was most conversant in. Participants were independently diagnosed as a case/non-case of dementia using the DSM-IV. Results—There were 259 participants (85 Chinese, 85 Malays, and 89 Indians, mean age 70.15 years, 65.4% female, 58.8% had no/minimal formal education); 22.8% (*n* = 59) had dementia. Correlations between the measures were substantial. All the measures had acceptable overall discriminative abilities. Diagnostic accuracies of the instruments did not differ across the ethnic groups. Effects of education were present in the cognitive screening measures. Conclusions—The CSI-D, CMMSE, and ECAQ are valid dementia instruments in this multi-ethnic Asian setting.

## 1. Introduction

The Global Burden of Disease (GBD) study estimated that there were 57.4 (95% uncertainty interval 50.4–65.1) million cases of dementia globally in 2019 [1]. The pooled prevalence of dementia in the World Health Organization’s (WHO) Southeast Asia Region (SEAR) was found to be 3% (95% confidence interval (CI) 2–6%) among those above the age of 60 years [2]. Rapid increases in numbers are expected with current trends in population ageing and population growth. Accurate disease burden estimates are key for informing policy, public health planning, and resource prioritisation.

Singapore, a city-state of 5.45 million [3], is situated in the heart of Southeast Asia. The multi-racial population comprises Chinese (74.3%), Malays (13.6%), Indians (8.9%), and others (3.2%). Residents have a life expectancy at birth in 2021 of 83.5 years; Singapore is among the most rapidly aging populations in the world, with 16% currently aged 65 years or above, estimated to reach 23.8% by 2030 [4]. While there is overall literacy rate of 97.6% with 11.3 mean years of schooling, the elderly generally have low educational attainments [5].

The influence of age, education, and ethnicity differences on dementia diagnosis are difficult issues that dementia researchers and clinicians grapple with, even more so in populations that are multi-racial, aging, with reduced literacy among the elderly, like in Singapore. The availability of evidence-based interventions makes it imperative that individuals with dementia are detected and diagnosed so that they may be adequately treated [6,7]. 

There are a number of screening tools for dementia [6,8]. These include the Abbreviated Mental Test (AMT), Addenbrooke’s Cognitive Examination Revised (ACE-R), Clock Drawing Test (CDT), Free and Cued Selective Reminding Test (FCSRT), Informant Questionnaire on Cognitive Impairment in the Elderly (IQCODE), Mattis Dementia Rating Scale (MDRS), Memory Impairment Scale (MIS), Mini-Cog, 7-Minute Screen (7-MS), Short Portable Mental Status Questionnaire (SPMSQ), and Telephone Interview for Cognitive Status (TICS). These tests have varying sensitivity and specificity likely due to clinical heterogeneity [8]. They also test slightly different domains, with some more limited in the number of domains tested than others [6].

Among these screening tools for dementia, the Mini Mental State Examination (MMSE) remains the most thoroughly studied instrument. Pooled estimates across 14 studies (*n* = 10,185) resulted in 88.3% sensitivity (95% CI, 81.3 to 92.9) and 86.2% specificity (95% CI, 81.8 to 89.7) for a cut-point of 23/24 or 24/25 to detect dementia [8]. The modified version of the MMSE, the Chinese Mini Mental State Examination (CMMSE) with age and education level cut-offs, is widely used to screen for dementia in Singapore, but it has only been validated among the Chinese [9]. The Elderly Cognitive Assessment Questionnaire (ECAQ), another widely used screening tool in Singapore, has only been validated among Chinese and Malays [10]. 

The 10/66 Dementia Research Group has recommended the Community Screening Instrument for Dementia (CSI-D) as the cognitive screening instrument in international studies of dementia [11]. This instrument is designed to capture the core elements in dementia diagnosis by using both cognitive testing of the patient and data from an informant. It was developed for use in populations with different educational, cultural, and linguistic backgrounds and has demonstrated good adaptability and utility in many dissimilar populations [12,13]. There is also a need for common standardised instruments with established validities across many cultures and settings as differences in study methods will limit the confidence with which variations in prevalence and incidence rates of dementia between different sites can be interpreted [14]. The CSI-D has already been validated in Chinese, Taiwanese, and Indian populations [15,16,17]. Thus, a validation of the CSI-D in Singapore is important as it may provide a potentially culture- and education-independent instrument for dementia screening purposes and also a much-needed tool for local and comparative epidemiological researchers in dementia. 

This study aimed to develop and validate locally suitable versions of the CSI-D, CMMSE, and ECAQ suitable for use in the Chinese, Malay, and Indian populations in Singapore. We also compared the utility of the three instruments for the diagnosis of dementia in this multi-ethnic population and explored the influences of education and ethnic differences on these instruments.

## 2. Materials and Methods

### 2.1. Participant Recruitment

All participants were recruited for this study from the pool of 15,000 (stratified by ethnicity in a Chinese/Malay/Indian ratio of 3:1:1) who participated in a community-based Stroke, Parkinson’s, EpilEpsy and Dementia in Singapore (SPEEDS) survey of neurological illness—they were aged 50 years and above, and resided in one of four residential areas whose demographic profile resembled the rest of the country. 

All those who were screened positive in Phase-I on a modified WHO screening tool for neurological disorders, and a proportion of those who were screened negative were assessed in Phase-II by a neurologist/geriatrician in a community ‘clinic’; a clinical diagnosis of dementia/no-dementia was made by the study clinician, using the Diagnostic and Statistical Manual of Mental Disorders (DSM)-IV criteria. To make the diagnosis, the clinician had to clearly establish, in an interview with the participant and caregiver, (1) the presence of amnesia, and either apraxia, aphasia, agnosia, or disturbed executive functioning and (2) that these impairments represented a decline from a previously higher level of functioning and were of sufficient severity to result in a loss of independence in either community functioning, home functioning, or self-care. Dementia severity was not rated as part of this survey; no other neurodiagnostic procedures were conducted. 

For the present study, the selection of cases was accomplished by screening Phase-II records using a random sampling procedure, for 90 subjects from each ethnic group of interest (Chinese, Malay and Tamil). Informed verbal consent to participate was obtained from either the participant or a caregiver over the telephone. Such consent was obtained from a total of 270 participant–caregiver dyads, with 90 from each ethnic group. 

The interviews were carried out by trained nurses in the participants’ residence. The Chinese and Malay participants were interviewed by a Chinese nurse fluent in English, Chinese, and Malay, while the Indian participants were interviewed by an Indian nurse fluent in English and Tamil. The interviewers were blind to the participants’ diagnosis at the time of the interview. The participants were all administered the CMMSE, ECAQ, and CSI-D in random order by the research nurses. 

### 2.2. Translation Procedures

The CSI-D consists of 2 parts—a 32-item test is administered to the participant to assess cognition across multiple domains without requiring reading ability, while a 26-item caregiver interview assesses the daily functioning and general health of the participant [11]. The CSI-D was translated into Mandarin and Malay, respectively, by bilingual local researchers fluent in both English and the target language and independently back-translated by other local investigators to English. The Tamil version was obtained from the 10/66 investigators in Chennai, India, and had been previously validated [17]. A panel of expert local reviewers, from the disciplines of neurology, psychiatry, psychology, and nursing, who were bilingual in English and either Chinese, Malay, or Tamil, assessed the instruments for their acceptability and their conceptual validity in the local context. Items judged inappropriate for the Singapore context were changed in accordance to the 10/66 Dementia Research Group’s guidelines. 

The CMMSE consists of 22 test items as two questions from the original MMSE, one pertaining to locality (city/county) and the other to season, were removed; the domains assessed included orientation, naming, arithmetic, recall, comprehension, and copying [9]. The ECAQ comprises 10 test items culled from the MMSE and Geriatric Mental State Schedule (GMSS)—it assesses orientation/information and memory [10]. Both scales were translated and back-translated for unavailable language versions and were similarly reviewed by the researchers. The items on both scales were considered appropriate for the local population and no changes were made. 

### 2.3. Data Analysis

Three summary scores were generated from the CSI ‘D’: (1) Cognitive Score (COGSCORE), an item-weighted score based on the participant’s performance on the cognitive component, (2) Informant Score (RELSCORE), an unweighted total score from the caregiver interview, and (3) Discriminate Function Score (DFSCORE), an item-weighted discriminant score combining COGSCORE and RELSCORE. The CMMSE yielded a single total score of maximum 28 while the ECAQ yielded a maximum score of 10.

All statistical analyses were conducted using the Statistical Package for Social Studies (SPSS) v19. Comparisons between the three ethnic groups were made on the demographic variables of age, gender, dementia status, and education level. Similar analyses for demographic differences between the participants with and without dementia were also carried out. Correlations were conducted between all the measures to investigate their convergent reliabilities while discriminative validity, sensitivity, and specificity were assessed using area under the receiver operator curve (ROC) as a measure of diagnostic accuracy. Descriptive analyses were conducted to assess for the roles of educational and ethnic differences.

## 3. Results

In total, 85 Chinese, 85 Malays, and 89 Indians completed all the required questionnaires; four had physical difficulties that prevented them from completing all the items (two Chinese, one Malay, and one Indian), six participants refused to complete all the questionnaires (three Chinese and three Malay), and a Malay participant did not have a caregiver available for the interview. The mean age of the study participants who completed the questionnaires was 70.15 years; 65.4% were female, 58.8% had no/minimal formal education, and 22.8% were demented.

The three ethnic groups did not differ significantly in terms of age, gender distribution, educational level, and cognitive status (Table 1). Overall, 77.2% participants (*n* = 200) were considered not to have dementia. As only Phase-II individuals were recruited, that is, those who had screened positive for neurological disease, a higher proportion of cases with dementia (when compared to community studies) was expected. There were no differences in gender distribution between those with and without dementia. However, significantly more dementia cases were older or had none/ minimal formal education (Table 1).

The correlations between all the measures were substantial (Table 2). However, the correlations between the RELSCORE and the cognitive measures (COGSCORE, CMMSE, ECAQ) were lower compared to the other correlations, suggesting that the RELSCORE taps a related but not identical, construct to the cognitive measures.

All five measures had acceptable overall diagnostic accuracies (Table 3). With the exception of the lower diagnostic accuracy for the RELSCORE for the Malays, which partially accounted for the lower overall diagnostic accuracy for the RELSCORE, the discriminative ability of the instruments to detect dementia generally did not differ across the different ethnic groups. The sensitivity and specificity of each instrument in this current sample are presented in Table 4. 

The mean scores on COGSCORE, CMMSE, and ECAQ did not differ significantly as a function of education for participants with dementia (Table 5). However, significant effects of education were present for those who were non-cases for dementia for all three instruments; the effect sizes (partial η^2^) associated with each instrument were 0.16, 0.16, and 0.07, for the COGSCORE, CMMSE, and ECAQ, respectively. There were no significant effects of education on the RELSCORE (partial η^2^ = 0.02), and the effect of education on the DFSCORE (partial η^2^ = 0.07), while significant, was reduced compared to the COGSCORE. 

There were significant effects of ethnicity on the COGSCORE (partial η^2^ = 0.13), CMMSE (partial η^2^ = 0.06), and ECAQ (partial η^2^ = 0.18) only for the non-demented participants, where the Indians had lower scores on average compared to the Chinese and Malays. The same pattern of effects was present for the RELSCORE (partial η^2^ = 0.11), where Indian caregivers reported more symptoms of decline in their non-demented elderly compared to the Chinese and Malays, as well as for the DFSCORE (partial η^2^ = 0.14). Although the differences were not considered significant, the mean RELSCORE for the Malays participants with dementia was notably lower compared to the Chinese and Indians.

## 4. Discussion

This study demonstrated that the CSI0D, CMMSE, and ECAQ are valid in a multi-ethnic, multilingual population for dementia screening purposes. Earlier dementia screening validation studies in Singapore focused only on the Chinese population; this study is one of the few to demonstrate the cross-cultural validity of these instruments in this population. Our study also went beyond mild AD and, for CSI-D, also included inputs from an informant for a more holistic evaluation of the patient. The overall diagnostic accuracies were acceptable and comparable for all the various measures (ranging from 0.85 for the RELSCORE to 0.98 for the COGSCORE) and, with the exception of the lower discriminative ability of the informant’s report for the Malays, the discriminatory ability of each measure was generally acceptable for each ethnic group. The instruments correlated highly with one another, indicating good convergent reliabilities between the instruments. However, the mean scores on the measures appeared to be influenced to some extent by education and ethnicity in this sample. 

Among the well-known dementia screening tests, the MMSE has a sensitivity of 88.3% and specificity of 86.2% [8]. The respective sensitivities and specificities of the ACE-R are 94% and 89%, AMT 42–100% and 83–95.4%, CDT 67–97.9% and 69–94.2%, IQCODE 75–87.6% and 65–91.1%, MDRS 98% and 97% (in Alzheimer’s disease), MC 76–100% and 54–85.2%, and MIS 43–86% and 93–97% [6,8].

The overall diagnostic accuracies of the CSI-D in this study are comparable to those reported by other studies [13,15] attesting to the adaptability of this instrument. The optimal cut-points in this sample are lower compared to the “official” cut-points (DFSCORE ≥ 0.184 and COGSCORE ≤ 28.5 for 100% sensitivity, 79% specificity) [18,19]. However, the associated sensitivity and specificity at a COGSCORE ≤ 28.5 is similar, which suggest that the difference for the discriminant score may be associated more with the informant’s report. Cultural influences on the informant’s report [13] and the lower diagnostic accuracy for the informant’s report for the Malay participants may account for the overall lower diagnostic accuracy for the RELSCORE and DFSCORE (further discussed below). The associated sensitivities and specificities for various cut-off points for the CMMSE are similar compared to those reported by Sahadevan et al. [9]. As indicated by Sahadevan et al., a cut-off of ≤5 for the ECAQ does appear to be overly stringent for detecting dementia, with the current study suggesting an optimal cut-off of 7. The lower level of education of the participants (85% had none to minimal education) in Kua and Ko’s study [10] and the higher proportion of dementia participants in this sample may be some reasons accounting for the difference.

The effects of education were found for all the cognitive measures. The effect size associated with education was similar for the COGSCORE and CMMSE while this effect was appreciably smaller for the ECAQ. This is unsurprising given that the cognitive requirements of the ECAQ are comparatively minimal. Similar to other studies [13,15], informant score was not significantly influenced by education. Consistent with Hall et al., [15] while the combination of cognitive score and informant score (DFSCORE) did not eliminate the education bias, it did reduce the size of the effect associated with this factor.

An effect of ethnicity was also present for all three cognitive measures in the participants without dementia—Indians achieved lower scores on average on these instruments though there were no ethnicity-based differences in discriminative ability for the cognitive instruments. This does not appear to be a result of differences in education as there were no significant differences between the three ethnic groups in terms of education level in this group (*p* = 0.99). The reasons underlying this difference are presently unclear. One possibility could be an inadvertent sampling of a greater number of Indians with mild cognitive impairment but who did not meet the criteria for dementia. The finding that Indian caregivers in this sample also tended to report more symptoms of cognitive decline is congruent with this possibility. However, significant effects of ethnicity remained for the COGSCORE (*p* = 0.01) and the ECAQ (*p* < 0.001), but not for the CMMSE (*p* = 0.43) after covarying for the RELSCORE. In contrast, when the variance associated with COGSCORE was accounted for, the differences in informant report (*p* = 0.09) and discriminant score (*p* = 0.09) were not considered significant. Understanding these findings would require a detailed item-by-item analysis of the various instruments, which is beyond the scope of this paper. Nevertheless, the influences of culture and language differences on different dimensions of cognitive functions as well as changes in cognitive functions over time are considered important issues to investigate [20] and further research into this is warranted. 

Another marginal influence of ethnicity was present in the diagnostic accuracies for the CSI-D informant report component where the diagnostic accuracy of the RELSCORE was notably lower for the Malays compared to the Chinese and Indians. Although the differences in the means were not considered significant, inspection of the means suggests that Malay caregivers may sometimes report fewer symptoms of cognitive and functional changes in their elderly with dementia compared to the Chinese and Indians. This may partially be a reflection of cultural and religious differences between how the different ethnic groups view and cope with illnesses. Cultural differences between the family environment (where there is a more extensive family network and better support system for the Malay elderly) and ability to cope with dementia in Chinese and Malay elderly had been discussed by Kua and Ko [21]. However, this may be changing with time due to the decreasing birth rates and increasing popularity of nuclear over extended families for all ethnic groups. 

Other dementia screening tests have been used in Singapore. A single question on progressive forgetfulness has only been validated in Chinese; similar with the Abbreviated Mental Test (AMT) and Frontal Assessment Battery (FAB) [9,22,23]. The Montreal Cognitive Assessment (MoCA) has only been validated in the clinic setting for mild Alzheimer’s disease (AD) [24]. The informant AD8 (iAD8) was superior to participant AD8 (pAD8), may be superior to MMSE, and is similar to MoCA [25,26,27]. A new test that has been developed in Singapore, the Visual Cognitive Assessment Test (vCAT), yielded similar accuracy as the MoCA, for mild cognitive impairment (MCI) and mild AD, without needing translation or adaptation, and thus would be applicable across cultures [28,29].

There are some limitations to this study. The overall prevalence of dementia was higher in the study population when compared with previous studies in Singapore [21,30,31,32,33,34,35]. However, this may be a reflection of the sampling procedure employed, with subjects included being drawn from Phase-II of a community-based survey, who were thus more likely to have dementia. This over-sampling of dementia cases, although inadvertent, was judged not to be necessarily disadvantageous, this being a study comparing dementia screening instruments, as opposed to a study of dementia prevalence. Further, the low prevalence of dementia in the community has posed problems for the development and validation of dementia instruments [36].

When adjusting for education and cognitive status, the number of dementia cases with higher levels of education becomes small. However, this is likely reflective of the relative prevalence of dementia in relation to educational level in the general population as the average level of educational attainment among the Singapore elderly is low [5]. There is also an increased risk of dementia associated with lower levels of education [37,38]. As a measure of dementia severity was not included in this study, further analysis of this data has not been possible. However, among the participants with dementia, there were no significant differences on their cognitive scores (COGSCORE, CMMSE, ECAQ) when comparisons were conducted for differences between the three races which suggests that to a certain extent, dementia severity may not have differed to a substantial degree between the three ethnic groups. As mentioned earlier, there may have been some inadvertent sampling of an increased number of Indians with mild cognitive impairment. While this may have partially contributed to the influence of ethnicity on the mean scores on the cognitive measures and informants’ report, as dementia diagnosis is a dichotomous decision, this does not in itself invalidate the findings that the instruments investigated in this study are valid dementia screening instruments in this population. Moreover, it should be kept in mind that these instruments are not meant to be used injudiciously as solitary dementia diagnostic tools, but rather, as screening measures that are part of a comprehensive dementia diagnostic package.

Given the reduced cognitive requirements and smaller education bias, the ECAQ may be more appropriate for use with elderly who have none to minimal education, while the CMMSE and the CSI-D may be more appropriate for those who have at least completed primary schooling. However, cognitive measures that are educationally biased also have a higher sensitivity to dementia as both characteristics are a result of a higher level of item difficulty [39]. As such, it will be valuable to investigate if instruments like the ECAQ can maintain high sensitivity with sufficient specificity, especially for mild dementia, in highly educated populations. Another method to reduce the education bias in cognitive testing may be to develop educationally adjusted cut-off scores^9^. Alternatively, the combination of cognitive score and the informant’s report can also effectively reduce the education bias and is closer to clinical practice [13].

Given the need for comparative methodology before meaningful comparisons of international (and national) differences of prevalence rates can be made, this development and validation of the CSI-D also provides a useful tool for dementia epidemiological studies, not just for Singapore, but also for other parts of Southeast Asia (e.g., Malaysia, Indonesia). Nevertheless, given the relative complexity in administration and scoring of the CSI-D, the CMMSE and ECAQ would be favoured in busy clinical settings. 

## 5. Conclusions

In summary, the CSI-D, CMMSE, and ECAQ are valid instruments for dementia screening purposes even in a multi-ethnic, multilingual setting like Singapore comprising ethnic Chinese, Malays, and Indians. The responses to these instruments may be influenced by factors such as education, as seen across all three screening tests, ethnicity (for example, the lower discriminative ability of the informant’s report for the Malays in the CSI-D), and religious and cultural beliefs (e.g., Indian caregivers reported more symptoms of decline in their non-demented elderly compared to the Chinese and Malays), in line with findings elsewhere. Future studies should be conducted to compare these instruments in a detailed item-by-item analysis, to further investigate the specificity and sensitivity of each item in relation to dementia severity and cognitive changes over time, and the influences of education, language, and culture on performance on these measures. A detailed item-by-item analysis would also contribute to the development of a shorter instrument suitable for clinical settings and the bedside, with the concurrent aim to minimise impacts of ethnicity and education. 

## Figures and Tables

**Table 1 healthcare-12-00410-t001:** The demographic information of the participants as a function of ethnicity and cognitive status.

		Ethnic Group				Statistic	*p*
		Chinese	Malay		Indian		
Age (yr)		M = 70.78	M = 70.68		M = 69.03	F(2, 256) = 1.73	0.18
		SD = 6.45	SD = 5.69		SD = 8.36		
Gender	Female	46	42		58	χ^2^(2) = 4.65	0.10
	Male	39	43		31		
Education Level	None/Minimal	51	54		50	χ^2^(2) = 0.98	0.61
	Primary/Secondary	34	31		39		
Cognitive Status	Dementia	21	20		18	χ^2^(2) = 0.54	0.77
	Normal	64	65		71		
		**Cognitive Status**				**Statistic**	** *p* **
		**Dementia**		**Normal**			
Age (yr)		M = 74.76		M = 68.79		t(257) = 6.17	0.00
		SD = 7.04		SD = 6.35			
Gender	Female	38		108		χ^2^(1) = 2.01	0.16
	Male	21		92			
Education Level	None/Minimal	51		104		χ^2^(1) = 22.49	0.00
	Primary/Secondary	8		96			

**Table 2 healthcare-12-00410-t002:** Correlations between the CSI ‘D’ variables, CMMSE, and ECAQ scores.

	COGSCORE	RELSCORE	DFSCORE	CMMSE
RELSCORE	−0.77	--		
DFSCORE	−0.91	0.97	--	
MMSE	0.94	−0.75	−0.87	--
ECAQ	0.89	−0.68	−0.81	0.86

Note: All correlations significant at *p* = 0.01.

**Table 3 healthcare-12-00410-t003:** Discriminatory ability (area under ROC curves with 95% confidence intervals) of each test by race.

	Overall	Chinese	Malay	Indian
COGSCORE	0.98 (0.96–0.99)	0.99 (0.97–1.00)	0.98 (0.95–1.00)	0.98 (0.96–1.00)
RELSCORE	0.85 (0.79–0.92)	0.94 (0.88–1.00)	0.73 (0.58–0.88)	0.95 (0.91–1.00)
DFSCORE	0.94 (0.91–0.97)	0.98 (0.96–1.00)	0.91 (0.85–0.97)	0.98 (0.96–1.00)
CMMSE	0.96 (0.93–0.98)	0.99 (0.97–1.00)	0.94 (0.88–1.00)	0.93 (0.88–0.98)
ECAQ	0.95 (0.92–0.98)	0.97 (0.92–1.00)	0.96 (0.92–1.00)	0.96 (0.93–1.00)

**Table 4 healthcare-12-00410-t004:** Sensitivity and specificity at different cut-off scores for the COGSCORE, RELSCORE, DFSCORE, CMMSE, and ECAQ in the current sample. The numbers shaded in gray indicate the most appropriate cut-offs for this sample while the numbers in italics indicate the “official” cut-offs that were established by previous studies [9,10,18,19]. Note that there are no established cut-off points for the RELSCORE.

	Cut-Off Points	Sensitivity	Specificity
COGSCORE	≤26.91	91.5	87.5
	≤27.40	96.6	85.5
	≤27.50	98.3	85.0
	≤27.80	100.0	83.5
	≤28.00	100.0	80.0
	≤*28.50*	*100.0*	*75.0*
RELSCORE	≥3.75	82.8	80.6
	≥4.25	81.0	83.1
	≥4.58	79.3	86.6
	≥4.83	79.3	87.1
	≥5.34	72.4	89.6
DFSCORE	≥0.0987	87.9	79.6
	≥0.1075	84.5	81.1
	≥0.1104	84.5	82.6
	≥0.1344	83.1	87.5
	≥0.1502	82.8	90.0
	≥*0.1843*	*77.6*	*93.5*
CMMSE	≤17.00	76.3	97.0
	≤18.00	83.1	94.5
	≤19.00	89.8	90.5
	≤*20.00*	*91.5*	*86.0*
	≤21.00	91.5	79.0
ECAQ	≤4.00	59.3	99.0
	≤*5.00*	*78.0*	*95.5*
	≤6.00	81.4	94.5
	≤7.00	89.8	91.5
	≤8.00	94.9	76.0

**Table 5 healthcare-12-00410-t005:** Means, standard deviations (in brackets), and 95% confidence intervals of COGSCORE, RELSCORE, DFSCORE, CMMSE, and ECAQ as a function of education level/race and cognitive status.

		Cognitive Status	
		Dementia	No Dementia
COGSCORE		21.49 (3.67); CI = 20.45–22.52	29.01 (2.58); CI = 28.52–29.52
Education	None/minimal	21.33 (4.06); CI = 20.19–22.48	29.01 (2.58); CI = 28.52–29.52
	1° or 2°	22.46 (3.38); CI = 19.63–25.29	30.97 (1.81); CI = 30.60–31.33
Race	Chinese	20.92 (4.85); CI = 18.71–23.13	30.72 (2.07); CI = 30.21–31.24
	Malay	22.84 (3.82); CI = 20.05–23.62	30.48 (2.54); CI = 29.85–31.11
	Indian	21.76 (3.02); CI = 20.26–23.26	28.78 (2.23); CI = 28.25–29.31
RELSCORE		7.69 (4.70); CI = 6.47–8.91	1.56 (2.26); CI = 1.25–1.88
Education	None/minimal	7.43 (4.87); CI = 6.06–8.80	1.86 (2.30); CI = 1.41–2.30
	1° or 2°	9.34 (3.15); CI = 6.71–11.98	1.24 (2.19); CI = 0.80–1.69
Race	Chinese	8.42 (4.25); CI = 6.48–10.36	0.86 (1.90); CI = 0.39–1.33
	Malay	5.74 (5.62); CI = 3.11–8.37	1.18 (2.26); CI = 0.63–1.74
	Indian	9.00 (3.41); CI = 7.31–10.70	2.54 (2.26); CI = 2.01–3.08
DFSCORE		0.33 (0.19); CI = 0.28–0.38	0.00 (0.10); CI = −0.01–0.01
Education	None/minimal	0.32 (0.19); CI = 0.27–0.38	0.02 (0.10); CI = 0.01–0.04
	1° or 2°	0.36 (0.15); CI = 0.24–0.48	−0.03 (0.09); CI = −0.05–−0.01
Race	Chinese	0.36 (0.18); CI = 0.28–0.44	−0.03 (0.08); CI = −0.06–−0.01
	Malay	0.26 (0.22); CI = 0.16–0.36	−0.02 (0.13); CI = −0.05–0.01
	Indian	0.36 (0.15); CI = 0.29–0.44	0.05 (0.09); CI = 0.03–0.07
CMMSE		14.66 (4.12); CI = 13.59–15.74	23.92 (3.01); CI = 23.50–24.34
Education	None/minimal	14.33 (4.12); CI = 13.17–15.49	22.76 (2.93); CI = 22.19–23.33
	1° or 2°	16.75 (3.65); CI = 13.69–19.81	25.18 (2.57); CI = 24.66–25.70
Race	Chinese	13.19 (4.13); CI = 11.31–15.07	24.70 (2.54); CI = 24.01–25.34
	Malay	14.70 (4.49); CI = 12.60–17.80	24.20 (3.25); CI = 23.39–25.01
	Indian	16.33 (3.12); CI = 14.78–17.89	22.96 (2.96); CI = 22.26–23.66
ECAQ		4.23 (1.98); CI = 3.91–4.94	8.87 (1.26); CI = 8.70–9.04
Education	None/minimal	4.29 (1.90); CI = 3.76–4.83	8.55 (1.44); CI = 8.27–8.83
	1° or 2°	5.25 (2.43); CI = 3.21–7.29	9.21 (0.93); CI = 9.02–9.40
Race	Chinese	4.52 (2.48); CI = 3.39–5.65	9.34 (0.80); CI = 9.14–9.54
	Malay	4.35 (1.98); CI = 3.42–5.28	9.17 (1.22); CI = 8.86–9.47
	Indian	4.39 (1.33); CI = 3.73–5.05	8.15 (1.34); CI = 7.84–8.47

## Data Availability

Data is available from Tan Tock Seng Hospital.

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
