# Peer review of "A Comparative Study of Three Dementia Screening Instruments (CSI-D, CMMSE, and ECAQ) in a Multi-Ethnic Asian Population"

_healthcare, 2024, doi:10.3390/healthcare12030410_

Round 1

Reviewer 1 Report

Comments and Suggestions for Authors

The manuscript describes a study to identify dementia in three major ethnic groups historically residing in Singapore. To do this, three small psychological tests are being tested to detect dementia in Singapore. In this regard, the work has both its pros and cons:

1. When developing and implementing new tests, it is traditional to use internationally valid psychological tests, adapted for countries and nationalities, for comparison. Here the author uses three tests that have fairly limited application. Probably only CMMSE is adapted. On this basis, the introduction requires an adequate overview of current international psychological diagnostic practices for dementia, leading the reader to an understanding of the use of the CSI-D, ECAQ and CMMSE.

2. In Translation Procedures: I recommend describing the tests in more detail. Since the tests are not large, it might make sense to place them in an Appendix to the manuscript. Or provide valid links to resources with these tests.

3. In the Discussion, it is also necessary to present the available information from the scientific literature on the study of dementia using well-known tests, followed by comparison with one’s own data.

4. The conclusion needs to be redone and made more specific, highlighting the most important own results for individual tests.

Author Response

Dear Reviewer 1

  1. When developing and implementing new tests, it is traditional to use internationally valid psychological tests, adapted for countries and nationalities, for comparison. Here the author uses three tests that have fairly limited application. Probably only CMMSE is adapted. On this basis, the introduction requires an adequate overview of current international psychological diagnostic practices for dementia, leading the reader to an understanding of the use of the CSI-D, ECAQ and CMMSE.

Response – thank you for this excellent suggestion. A new paragraph has now been added to the Introduction (lines 49-57):

There are a number of screening tools for dementia [6,8]. These include the Abbreviated Mental Test (AMT), Addenbrooke’s Cognitive Examination Revised (ACE-R), Clock Drawing Test (CDT), Free and Cued Selective Reminding Test (FCSRT), Mattis Dementia Rating Scale (MDRS), Memory Impairment Scale (MIS), Mini-Cog, 7-Minute Screen, Short Portable Mental Status Questionnaire (SPMSQ), Telephone Interview for Cognitive Status (TICS), and Informant Questionnaire on Cognitive Impairment in the Elderly (IQCODE). These tests have varying sensitivity and specificity likely due to clinical heterogeneity [8]. They also test slightly different domains, with some more limited in the number of domains tested than others [6].

  1. In Translation Procedures: I recommend describing the tests in more detail. Since the tests are not large, it might make sense to place them in an Appendix to the manuscript. Or provide valid links to resources with these tests.

Response – thank you for this kind advice. Unfortunately, due to copyright issues, I am unable to reproduce the tests and place them in the paper, even as an appendix. However, a brief description of each test has now been added, and the references provided for the interested reader:

The CSI-D consists of 2 parts - a 32-item test is administered to the participant to assess cognition across multiple domains without requiring reading ability, while a 26-item caregiver interview assesses the daily functioning and general health of the participant [11] (lines 123-126).

The CMMSE consists of 22 test items as two questions from the original MMSE, one pertaining to locality (city/county) and the other to season, were removed – the domains assessed include orientation, naming, arithmetic, recall, comprehension and copying [9] (lines 137-139).

The ECAQ comprises 10 test items culled from the MMSE and Geriatric Mental State Schedule – it assesses orientation/information and memory [10] (lines140-142).

  1. In the Discussion, it is also necessary to present the available information from the scientific literature on the study of dementia using well-known tests, followed by comparison with one’s own data.

Response – thank you for this wonderful recommendation. A paragraph has now been added to Discussion giving the sensitivity and specificity of the well-known tests before the paragraph with this study’s data (lines 243-247):

Among the well-known dementia screening tests, the MMSE has a sensitivity of 88.3% and specificity of 86.2% [8]. The respective sensitivities and specificities of the ACE-R are 94% and 89%, AMT 42-100% and 83-95.4%, CDT 67-97.9% and 69-94.2%, IQCODE 75-87.6% and 65-91.1%, MDRS 98% and 97% (in Alzheimer’s Disease), Mini-Cog 76-100% and 54-85.2%, and MIS 43-86% and 93-97% [6,8].

  1. The conclusion needs to be redone and made more specific, highlighting the most important own results for individual tests.

Response - I apologise for not being more specific. The opening sentences of the Conclusion has now been amended (lines 365-371):

In summary, CSI-D, CMMSE, and ECAQ are valid instruments for dementia screening purposes even in a multi-ethnic, multilingual setting like Singapore comprising ethnic Chinese, Malays and Indians. The responses to these instruments may be influenced by factors such as education as seen across all 3 screening tests, ethnicity (for example the lower discriminative ability of informant’s report for the Malays in the CSI-D), religious and cultural beliefs (eg. Indian caregivers reported more symptoms of decline in their non-demented elderly compared to the Chinese and Malays).

Thank you very much

Reviewer 2 Report

Comments and Suggestions for Authors

This manuscript describes the comparison of different dementia screening tools for multi-ethnic Asian populations comprising Chinese, Malays and Indians in Singapore. The author found out that all measures had acceptable overall discriminative abilities, and all are valid dementia instruments in this multi-ethnic Asian setting.

A few concerns to the authors.

1.       Some abbreviations will need to be explained with the full description for the first time. It is also very helpful to have all the abbreviations and their full description listed at the end of the manuscript.

2.       A little bit more information about all the measures like CMMSE, ECAQ, CSI-D, COGSCORE, and RELSCORE and DEFSCORE will be needed. Are they self-reporting questionnaires or evaluations from healthcare professionals?

3.       In table 4, how can sensitivity and specificity be determined?

Author Response

Dear Reviewer 2

  1. Some abbreviations will need to be explained with the full description for the first time. It is also very helpful to have all the abbreviations and their full description listed at the end of the manuscript.

Response – I apologise for this omission. All abbreviations are now fully described when used for the first time. All the abbreviations and their full description are now listed at the end of the manuscript (lines 380-412):

AMT - Abbreviated Mental Test

ACE-R - Addenbrooke’s Cognitive Examination-Revised

AD – Alzheimer’s Disease

CDT - Clock Drawing Test

CMMSE - Chinese Mini Mental State Examination

COG-SCORE - Cognitive Score

CSI-D - Community Screening Instrument for Dementia

DEFSCORE - Discriminate Function Score

DSM - Diagnostic and Statistical Manual of Mental Disorders

ECAQ - Elderly Cognitive Assessment Questionnaire

FCSRT - Free and Cued Selective Reminding

FAB - Frontal Assessment Battery

GMSS - Geriatric Mental State Schedule

IQCODE - Informant Questionnaire on Cognitive Impairment in the Elderly

GBD - Global Burden of Disease

MC- Mini-Cog

MDRS - Mattis Dementia Rating Scale

MIS - Memory Impairment Scale (MIS)

MMSE - Mini Mental State Examination

MoCA - Montreal Cognitive Assessment

p - probability

RELSCORE - Informant Score

ROC - Receiver Operator Curve

7-MS - 7-Minute Screen

SEAR - Southeast Asia Region

SPEEDS - Stroke, Parkinson’s, EpilEpsy and Dementia in Singapore

SPMSQ - Short Portable Mental Status Questionnaire

SPSS – Statistical Package for Social Studies

TICS - Telephone Interview for Cognitive Status

vCAT - Visual Cognitive Assessment Test

WHO – World Health Organisation

  1. A little bit more information about all the measures like CMMSE, ECAQ, CSI-D, COGSCORE, and RELSCORE and DEFSCORE will be needed. Are they self-reporting questionnaires or evaluations from healthcare professionals?

Response – the CMMSE, ECAQ and CSI-D are questionnaire-based tests administered by the healthcare professionals, while COGSCORE, RELSCORE and DEFSCORE are calculated from the responses on the CSI-D. That the CMMSE, ECAQ and CSID were administered by the research nurses is now clarified in the paper (lines 119-120).

The participants were all administered the CMMSE, ECAQ and CSI-D in random order by the research nurses.

The generation of the scores was previously mentioned (now lines 148-153).

  1. In table 4, how can sensitivity and specificity be determined?

Response – I apologise for not mentioning this - these were derived via the ROC curves – this information is now mentioned in the Methods (lines 159-161):

Correlations were conducted between all measures to investigate their convergent reliabilities while discriminative validity, sensitivity and specificity was assessed using area under the Receiver Operator Curve (ROC) curve as a measure of diagnostic accuracy.

Thank you very much

Round 2

Reviewer 1 Report

Comments and Suggestions for Authors

In its present form, the manuscript looks much more interesting and can be recommended for publication.